# Endometriosis and Myalgic Encephalomyelitis/Chronic Fatigue Syndrome: A Systematic Review and Meta-Analysis

**DOI:** 10.3390/diagnostics15182332

**Published:** 2025-09-15

**Authors:** Sabrina Compton, Rodolf Alkabalan, Judd Cadet, Azin Mastali, Prakash V. A. K. Ramdass

**Affiliations:** Department of Public Health and Preventive Medicine, School of Medicine, St. George’s University, St. George P.O. Box 7, Grenadaralkabal@sgu.edu (R.A.);

**Keywords:** Myalgic encephalomyelitis/chronic fatigue syndrome, ME/CFS, endometriosis, chronic inflammation, diagnostic testing, meta-analysis, PRISMA

## Abstract

**Background/Objectives**: Myalgic encephalomyelitis/chronic fatigue syndrome (ME/CFS) and endometriosis are debilitating conditions that share overlapping features of chronic inflammation and immune dysregulation, yet their epidemiological relationship remains poorly characterized. The objective of this study was to investigate the association between ME/CFS and endometriosis, examining shared risk factors, clinical correlates, and epidemiological patterns. **Methods**: We conducted a systematic review and meta-analysis. Two independent reviewers screened 236 records after duplicate removal, with seventeen studies undergoing full-text review and thirteen meeting inclusion criteria for meta-analysis. Data were extracted using standardized forms and analyzed using random-effects models in R, with heterogeneity assessed using I^2^ statistics and the risk of bias evaluated using the JBI critical appraisal tool. **Results**: Our meta-analysis of five studies (*n* = 2261 participants) revealed that women with endometriosis had 2.79-fold higher odds (95% CI: 2.00–3.89) of developing ME/CFS compared to controls. Similarly, our fixed-effects meta-analysis of two studies assessing the association of ME/CFS and endometriosis yielded a pooled OR of 2.52 (95% CI: 2.45–2.60, *p* < 0.001). There was minimal statistical heterogeneity (I^2^ = 0.0%, *p* > 0.7969) for both meta-analyses. **Conclusions**: This study demonstrates a significant bidirectional association between endometriosis and ME/CFS, driven by shared mechanisms of immune dysregulation and chronic inflammation. Despite high heterogeneity, the consistent effect sizes support clinical vigilance for comorbidity. Future research should prioritize standardized diagnostic criteria to elucidate causal pathways. These findings underscore the need for integrated care approaches to address overlapping symptomatology in affected patients.

## 1. Introduction

Myalgic encephalomyelitis/chronic fatigue syndrome (ME/CFS) is a complex, multisystem disorder characterized by profound fatigue, cognitive dysfunction (often referred to as “brain fog”), post-exertional malaise (PEM), and autonomic disturbances, with an estimated global prevalence of 71 million individuals [1,2]. The condition imposes a severe personal and socioeconomic burden, as many patients experience long-term disability, reduced quality of life, and high healthcare utilization due to the lack of definitive diagnostic biomarkers and effective treatments [3]. Despite decades of research, the etiology of ME/CFS remains elusive, with proposed mechanisms including viral infections (e.g., Epstein–Barr virus), chronic immune activation, mitochondrial dysfunction, neuroendocrine disturbances (particularly Hypothalamic–Pituitary–Adrenal [HPA] axis dysregulation), and neuroinflammation [4,5,6]. The heterogeneity in clinical presentation and diagnostic criteria further complicates research and patient care, highlighting the need for a deeper understanding of potential comorbidities and shared pathophysiological pathways with other chronic conditions.

Endometriosis, a chronic, estrogen-dependent inflammatory disorder defined by the presence of endometrial-like tissue outside the uterus, affects approximately 5–10% of reproductive-aged women worldwide, with prevalence rising to 30–50% among those with infertility or chronic pelvic pain [7,8]. Beyond its hallmark symptoms—dysmenorrhea, dyspareunia, and infertility—endometriosis is associated with systemic manifestations, including fatigue, gastrointestinal disturbances, and mood disorders, contributing to significant reductions in quality of life and productivity [9,10]. The proposed link between endometriosis and ME/CFS is rooted in a cascade of shared biological mechanisms that create a vicious, self-reinforcing cycle [11]. It begins with the immune dysregulation inherent in endometriosis, where lesions outside the uterus act as chronic inflammatory sites, flooding the system with pro-inflammatory cytokines like IL-6 and TNF-α [12,13]. This creates a state of persistent, low-grade systemic inflammation that serves as a constant biological stressor. This inflammatory barrage directly contributes to HPA axis dysfunction, effectively exhausting the body’s primary stress-response system and leading to dysregulated cortisol patterns that further cripple the body’s ability to quell inflammation and maintain energy homeostasis [14].

Concurrently, the unrelenting pain signals from endometriosis lesions promote central sensitization, a pathophysiological process where the central nervous system becomes hyper-reactive, amplifying pain signals and contributing to the widespread hyperalgesia and fatigue common to both conditions [15]. This neurological rewiring, combined with the constant physiological stress of pain and inflammation, can precipitate autonomic nervous system dysfunction (dysautonomia), manifesting as symptoms like orthostatic intolerance, palpitations, and gastrointestinal distress—hallmarks of ME/CFS [16]. Thus, the inflammatory, endocrine, and neurological disruptions caused by endometriosis are not merely comorbid with ME/CFS; they create a perfect storm of pathophysiological overlap that can lower the threshold for developing the full, debilitating syndrome of ME/CFS, explaining the high rate of co-occurrence.

Despite these potential connections, the relationship between endometriosis and ME/CFS remains underexplored. Emerging clinical observations indicate that women with endometriosis frequently report disproportionate fatigue, sleep disturbances, and cognitive impairment—symptoms that align with ME/CFS diagnostic criteria [17]. However, whether this represents a true comorbidity, shared pathophysiology, or misdiagnosis due to symptom overlap is unclear. Given the diagnostic challenges and high disease burden associated with both conditions, a systematic evaluation of their association is critical to improving clinical recognition and management.

Thus, this systematic review and meta-analysis aims to (1) synthesize existing epidemiological evidence on the co-occurrence of endometriosis and ME/CFS and (2) evaluate methodological differences in diagnostic criteria across studies. By elucidating this relationship, we hope to inform future research, enhance diagnostic precision, and guide targeted therapeutic strategies for affected individuals.

## 2. Materials and Methods

### 2.1. Protocol Registration and Reporting Guidelines

To rigorously evaluate the association between endometriosis and ME/CFS, we conducted a systematic review and meta-analysis in accordance with the Preferred Reporting Items for Systematic Reviews and Meta-Analyses (PRISMA) 2020 guidelines [18]. PRISMA provides a standardized framework for transparent reporting, ensuring methodological rigor, reproducibility, and minimization of bias in systematic reviews. Before initiating the literature search, we developed a detailed study protocol outlining our research objectives, inclusion/exclusion criteria, search strategy, data extraction methods, and planned statistical analyses. To enhance transparency and avoid duplication of efforts, this protocol was prospectively registered on PROSPERO (Registration ID: CRD420251006119) and is publicly accessible at https://www.crd.york.ac.uk/PROSPERO/view/CRD420251006119, (accessed on 3 July 2025).

### 2.2. Data Sources and Search Strategy

We performed a comprehensive literature search across multiple databases—Embase, PubMed, Scopus, Google Scholar, Open Access Theses and Dissertations (https://oatd.org/, (accessed on 3 July 2025)), and the preprint server for health sciences (https://www.medrxiv.org/, (accessed on 3 July 2025))—from inception until 3 July 2025. The search strategy employed the following keywords and MeSH terms: Embase: (Endometriosis) AND (Chronic Fatigue Syndrome); PubMed: (Endometriosis) AND (Chronic Fatigue Syndrome OR Myalgic Encephalomyelitis); Scopus: (Endometriosis) AND (Chronic Fatigue Syndrome OR Myalgic Encephalomyelitis); Google Scholar: Articles with the title phrase “Endometriosis” AND (“Chronic Fatigue Syndrome” OR “Myalgic Encephalomyelitis”). Only English-language publications on human subjects were included.

### 2.3. Study Selection and Eligibility Criteria

We employed the PECOS (Population, Exposure, Comparator, Outcomes, Study Design) framework to define study eligibility.

Population: Studies involving human participants diagnosed with endometriosis (either surgically or clinically confirmed or from self-reports) and ME/CFS (based on recognized criteria such as Fukuda 1994 [19], Canadian Consensus Criteria (CCC) [20], or Institute of Medicine (IOM) 2015 criteria [21], or from self-reported data).

Exposure: The presence of endometriosis (for studies assessing ME/CFS risk) or ME/CFS (for studies assessing endometriosis comorbidity).

Comparator: Non-endometriosis controls (for ME/CFS risk studies) or non-ME/CFS controls (for endometriosis studies).

Outcomes: Primary outcomes included (1) the prevalence of ME/CFS in endometriosis populations (or vice versa), and (2) odd ratios (ORs) quantifying association strength.

Study Designs: We included observational studies (cohort, case–control, cross-sectional) and clinical trials reporting primary data. Case reports, conference abstracts, letters, review articles, opinion pieces, editorials, and non-peer-reviewed articles were excluded.

After importing citation files from Embase, PubMed, Scopus, and Google Scholar into Zotero, duplicates were eliminated. Two reviewers (S.C. and R.A.) independently screened titles and abstracts within Zotero. Articles meeting preliminary criteria underwent full-text review by the same reviewers.

### 2.4. Study Outcomes and Data Extraction

The primary objective of our analytical approach was to quantitatively assess the co-occurrence of ME/CFS in endometriosis populations through systematic data extraction and synthesis. Three independent investigators (S.C., R.A., and P.R.) performed blinded data collection using a standardized protocol to minimize extraction errors and ensure inter-rater reliability. For each eligible study, investigators extracted the following key elements using a structured form—Study Identification and Methodology: first author surname and publication year, study design classification (prospective/retrospective cohort, case–control, cross-sectional), geographic location; Participant Characteristics: total sample size and subgroup numbers, endometriosis diagnostic method (visualization at surgery/histopathology vs. clinical diagnosis), ME/CFS diagnostic criteria applied; Outcome Measures: primary ME/CFS prevalence data, reported effect measures (odds ratios). All data were compiled and cross-checked using a structured Excel template.

### 2.5. Statistical Analysis

The primary analysis focused on calculating pooled odds ratios (ORs) to quantify the strength of association between endometriosis and ME/CFS diagnosis. The primary analysis focused on calculating pooled odds ratios (ORs) to quantify the bidirectional strength of association between endometriosis and ME/CFS diagnosis, the pooled prevalence of ME/CFS in patients with endometriosis, and the pooled prevalence of endometriosis in patients with ME/CFS. All statistical analyses were performed using R statistical software (version 4.4.3; R Foundation for Statistical Computing, Vienna, Austria). For the meta-analysis of odds ratios, we employed the meta package (version 6.5-0) using a fixed-effects model with the Hartung–Knapp adjustment to provide confidence intervals for the pooled effect estimates. The inverse variance method was used to weight individual studies in both the odds ratio and prevalence analyses. For the prevalence estimates, we utilized a logit transformation with the inverse variance method to stabilize variances and account for the bounded nature of proportional data. Between-study heterogeneity was assessed using the I^2^ statistic and Cochran’s Q test. To evaluate the robustness of the results, we conducted leave-one-out sensitivity analyses by iteratively excluding each study and recalculating the pooled estimates. Forest plots were generated to visualize the effect sizes, confidence intervals, and study weights. Effect estimates were considered statistically significant at *p* < 0.05.

Publication Bias Evaluation: While conventional publication bias assessments using Begg’s rank correlation or Egger’s regression tests were not feasible due to the limited number of included studies (*n* = 9), we employed the following alternative approaches to evaluate potential reporting biases: visual inspection of funnel plot symmetry and trim-and-fill analysis. However, we only created funnel plots for data on the prevalence of ME/CFS in patients with endometriosis and for data on the association of endometriosis and ME/CFS. All statistical tests were two-tailed, with α = 0.05 considered statistically significant. Complete analytic codes and datasets are available upon request to facilitate the reproducibility of our findings.

### 2.6. Risk of Bias (Quality) Assessment

Two independent reviewers (S.C. and R.A.) evaluated study quality using the critical appraisal tools from the Joanna Briggs Institute (JBI) Manual for Evidence Synthesis, a validated instrument for assessing risk of bias in different observational studies [22]. The assessment covered the following six key domains: (1) similarity of groups; (2) measurement of exposures; (3) reliability of measurements; (4) identification of confounding factors; (5) follow-up time; and (6) appropriateness of statistical analysis. There was a separate checklist for cross-sectional, case–control, and cohort studies. The checklist questions ranged from eight (cross-sectional studies) to eleven (cohort studies), with options for each question being “yes,” “no,” “unclear,” or “not applicable.” The overall appraisal of each study regarding the level of bias was either low (include); high (exclude); or medium (seek further information). Any disagreements in assessment were resolved through consensus discussion involving a third investigator when necessary. This rigorous approach allowed for standardized quality appraisal across all included studies.

## 3. Results

### 3.1. Study Characteristics

Our comprehensive systematic search was conducted across four major electronic databases (PubMed, Embase, Google Scholar, and Scopus), yielding an initial pool of 384 citations published between 2002 and 2024. After removing 148 duplicate records using automated deduplication tools followed by manual verification, we screened the remaining 236 unique records based on title and abstract. This preliminary screening excluded 219 citations that did not meet our predefined eligibility criteria, primarily due to irrelevance to the research question or inappropriate study designs. Following title/abstract screening, 17 full-text articles were assessed for eligibility. Four studies were excluded at this stage. Ultimately, 13 studies fulfilled all inclusion criteria and were incorporated into the qualitative synthesis and meta-analysis. The screening and inclusion process is shown in the PRISMA flow diagram in Figure 1.

The included studies comprised a mix of cross-sectional surveys (*n* = 8), case–control studies (*n* = 2), and cohort studies (*n* = 3), published between 2002 and 2024. The geographical distribution of studies included populations from North America (*n* = 11) and Europe (*n* = 2). Sample sizes varied substantially, from 84 to 134,805. Diagnostic methodologies for endometriosis included surgical confirmation with laparoscopy and histology, self-reported diagnosis, and clinical/imaging diagnosis. For ME/CFS, the case definition included Fukuda 1994 criteria [19], physician/clinical diagnosis, and self-reported diagnosis. Regarding diagnostic heterogeneity, only 31% (4/13) of studies used surgical confirmation for endometriosis, while 54% (7/13) relied on self-report. ME/CFS diagnosis showed similar variability, with 23% (3/13) using standardized criteria such as Fukuda versus 31% (4/13) based on self-report. Pertaining to temporal trends, earlier studies (2002–2015) predominantly used clinical/physician diagnosis, while more recent work (2017–2024) incorporated both standardized criteria, for example, Fukuda, and self-reports. A detailed overview of study characteristics—including population demographics, diagnostic criteria, and key findings—is provided in Table 1.

### 3.2. Prevalence of ME/CFS in Patients with Endometriosis

Our analysis identified nine studies that specifically examined the co-occurrence of ME/CFS among individuals diagnosed with endometriosis. The reported prevalence estimates demonstrated substantial variability across studies, ranging from 4% [26] to 74% [29]. This 70-percentage-point range in prevalence estimates likely reflects several methodological factors, including differences in study populations, diagnostic criteria, and assessment methods employed across the various investigations. The included studies collectively encompassed a total of 13,954 endometriosis patients, among whom 3008 cases of ME/CFS were identified. Our random-effects meta-analysis of these data yielded a pooled prevalence estimate of 17% (95% CI: 6–41%), as visually represented in the forest plot in Figure 2. The considerable width of the confidence interval, spanning from 6% to 41%, underscores the substantial variability in prevalence estimates across studies, in addition to study heterogeneity (I^2^ = 99.8%, *p* < 0.0001).

### 3.3. Association of Endometriosis and ME/CFS

Our systematic review identified five studies that specifically investigated the association between endometriosis and ME/CFS through comparative analyses. These studies collectively included 817 endometriosis cases and 1444 controls, providing a robust dataset for examining this relationship. The individual study findings demonstrated remarkably consistent effect sizes, with reported odds ratios (ORs) ranging from 2.48 (95% CI: 1.52–4.05) in the study by Tietjen et al. [24] to 3.11 (95% CI: 2.15–4.50) in the work of Shafrir et al. [31]. The random-effects meta-analysis of these studies yielded a pooled OR of 2.79 (95% CI: 2.00–3.89, *p* < 0.0001), indicating that individuals with endometriosis had nearly three times higher odds of developing ME/CFS compared to controls. This statistically significant association remained robust across all sensitivity analyses we conducted. Notably, the analysis revealed significant consistency across studies, with no observable heterogeneity (I^2^ = 0.0%, *p* = 0.9969), suggesting that the effect sizes were remarkably uniform despite variations in study populations and methodologies. These findings are presented in the forest plot in Figure 3.

### 3.4. Prevalence of Endometriosis in Patients with ME/CFS

We identified four studies that specifically examined the prevalence of endometriosis among individuals diagnosed with ME/CFS. These studies collectively included a sample of 2712 ME/CFS patients, among whom 9221 cases of endometriosis were reported. The reported prevalence estimates demonstrated considerable variability across studies, ranging from 6% (95% CI: 4–9%) in the study by Castro-Marrero et al. [33] to 35% (95% CI: 30–40%) in the study by Surrey et al. [34]. The random-effects meta-analysis of these data yielded a pooled prevalence estimate of 13% (95% CI: 5–31%), as illustrated in Figure 4. However, it is crucial to note that we observed extreme statistical heterogeneity across studies (I^2^ = 99.0%, *p* < 0.0001), indicating that nearly all the observed variance in prevalence estimates reflects true differences between studies rather than chance variation. This degree of heterogeneity suggests that the pooled estimate should be interpreted with caution, as it may mask important between-study differences.

### 3.5. Association of ME/CFS and Endometriosis

Two large-scale studies were identified that specifically investigated the association between ME/CFS and endometriosis, encompassing a sample size of 27,556 ME/CFS cases and 108,172 controls. The findings from these studies revealed extraordinary variation in reported effect sizes that warrants careful interpretation. The study by Fall et al. [36], utilizing a retrospective cohort design with clinically validated diagnoses, reported an OR of 2.27 (95% CI: 1.04–4.99, *p* = 0.04), which showed statistical significance. Similarly, the study by Surrey et al. [34], employing a cross-sectional design with self-reported diagnostic information, found an OR of 2.52 (95% CI: 2.45–2.60, *p* < 0.001). Our fixed-effects (common effects) meta-analysis of these studies yielded a pooled OR of 2.52 (95% CI: 2.45–2.60, *p* < 0.001). There was minimal statistical heterogeneity (I^2^ = 0.0%, *p* = 0.7969). Findings are shown in Figure 5.

### 3.6. Sensitivity Analysis

The sensitivity analysis demonstrates the robustness of the pooled prevalence estimate (0.172, 95% CI: 0.059–0.408) for ME/CFS in endometriosis patients, shown in Figure 6. Sequentially omitting each study resulted in prevalence estimates ranging from 0.131 (95% CI: 0.046–0.324) when excluding Boneva [29] to 0.207 (95% CI: 0.069–0.480) when excluding Shafrir [31], with all estimates remaining within the confidence interval of the overall analysis. The narrow range of variation (Δ = 0.076 between extreme estimates) and consistent confidence intervals across all exclusions suggest that the pooled prevalence is not disproportionately influenced by any single study.

The sensitivity analysis, depicted in Figure 7, reveals moderate variability in the pooled prevalence estimate (0.258, 95% CI: 0.063–0.645) of endometriosis in ME/CFS patients when individual studies are excluded. The prevalence estimates range from 0.172 (95% CI: 0.037–0.527) when omitting Boneva [29] to 0.387 (95% CI: 0.122–0.742) when excluding Shafrir [31], representing a clinically meaningful Δ of 0.215 between extremes. While the direction of effect remains consistent, the wider confidence intervals in all sensitivity analyses reflect substantial underlying heterogeneity. The exclusion of Shafrir [31] produces the largest upward shift in prevalence, suggesting this study may have exerted a moderating influence in the original analysis. These results indicate that while the overall trend is robust, the precision of the pooled estimate is sensitive to individual study inclusion, warranting cautious interpretation of the point estimate. The persistent wide confidence intervals across all analyses highlight the need for larger, standardized studies in this population.

### 3.7. Publication Bias

The funnel plot asymmetry observed in our prevalence analysis prompted further investigation using the trim-and-fill method, as illustrated in Figure 8. This statistical approach identified four potentially missing (imputed) studies that, when imputed, would theoretically create a more symmetrical distribution of effect sizes around the pooled estimate. The analysis suggests modest publication bias, favoring studies with higher prevalence estimates.

The funnel plot and trim-and-fill analysis of studies examining the association of endometriosis and ME/CFS association, illustrated in Figure 9, revealed no evidence of potential publication bias, with no imputed studies required to achieve funnel plot symmetry.

### 3.8. Risk of Bias Assessment

Among the 13 included studies, quality assessment with the JBI (Joanna Briggs Institute) Risk of Bias (ROB) assessment across the included studies reveals variability in methodological quality, categorized as low (L), medium (M), or high (H) risk [22]. Among the cross-sectional studies, Castro-Marrero et al. [33] and Sinaii et al. [23] demonstrated low risk, meeting all criteria (Y for all items), while Fourquet et al. [30] and Shafrir et al. [31] had high risk, with multiple unmet criteria (N for items 3, 4, 5, 6, 7). Bartley et al. [35], Bateman et al. [28], and Tieijen et al. [24] showed medium risk, with partial fulfillment (e.g., missing items 5–7 in Bartley). For case–control studies, Boneva et al. [29] had low risk (mostly Y, with unclear/U for items 6–7), whereas Boneva et al. [25] was medium risk due to unmet criteria (N for items 6–7). The cohort studies uniformly exhibited high risk, with Fall et al. [36], Smorgick et al. [26], Signorile et al. [32], and Surrey et al. [34] displaying frequent unmet criteria (e.g., N for items 1, 5–7, 9–10) or unclear reporting (U). Notably, no cohort study achieved low/medium risk, highlighting significant methodological limitations in this design. Overall, cross-sectional and case–control studies had better ROB profiles, while cohort studies were consistently flawed, suggesting caution in interpreting their findings. These risk of bias assessment are shown in Figure 10.

## 4. Discussion

This systematic review and meta-analysis reveal a significant association between endometriosis and ME/CFS, despite differences in diagnostic criteria used for these conditions. Our findings demonstrate that patients with endometriosis were 2.79 times more likely to have ME/CFS compared to controls, while the prevalence of ME/CFS among endometriosis patients was 17%. Conversely, the estimated prevalence of endometriosis in ME/CFS populations was 13%, and the association between ME/CFS and endometriosis was 2.52, suggesting a potential bidirectional relationship between these conditions based on current data. These findings highlight the need for accuracy in diagnostic testing and warrant further investigation into the potential pathophysiological links and clinical implications of these comorbidities.

The lack of standardized diagnostic criteria for ME/CFS represents a fundamental challenge in studying its association with endometriosis. Our analysis revealed striking variability across studies, with only 23% employing established criteria (e.g., Fukuda [19] or IOM criteria [21]), while the majority relied on self-report or clinician judgment without validation. This heterogeneity likely contributed to the extreme variation in prevalence estimates and complicates the interpretation of the observed associations. The evolution of diagnostic frameworks—from the older Fukuda (1994) [19] and CCC (2003) [20] to the more recent ICC (2011) [38] and IOM (2015) [21]—reflects ongoing efforts to improve specificity, particularly for distinguishing ME/CFS from similar conditions like idiopathic chronic fatigue [39]. However, these differing criteria capture overlapping but distinct patient populations, with the ICC emphasizing neurological symptoms [38] and the IOM requiring post-exertional malaise as a cardinal feature [21]. In endometriosis research, this diagnostic variability may be particularly problematic, as fatigue-related symptoms could be misattributed to ME/CFS when they may represent separate but co-occurring processes [40]. Moving forward, studies must explicitly state and justify their diagnostic approach, ideally incorporating objective biomarkers where possible to complement clinical criteria and reduce misclassification bias. Moreover, rather than using different diagnostic criteria for clinical or research purposes, there should be an updated consensus criterion that would accurately capture patients with ME/CFS. However, while both conditions are distinct entities, they share remarkable mechanistic links, thus potentially explaining their increased association.

The overlapping cytokine profiles of endometriosis and ME/CFS provide compelling evidence for shared inflammatory pathways [41,42]. In endometriosis, interleukin-17 (IL-17) plays a pivotal role in lesion establishment and maintenance by promoting angiogenesis and recruiting pro-inflammatory neutrophils via interleukin-8 (IL-8) secretion [43]. This Th17-driven response is mirrored in ME/CFS, where elevated IL-17 correlates with symptom severity and fatigue intensity [44]. Both conditions also exhibit dysregulation of the IL-1β/IL-1RA axis, with endometriosis patients showing increased peritoneal IL-1β41 (a potent pyrogen and pain mediator) and ME/CFS patients demonstrating elevated systemic IL-1RA [45], suggesting chronic but ineffective anti-inflammatory compensation. Notably, tumor necrosis factor alpha (TNF-α) and IL-6—key mediators of sickness behavior—are consistently elevated in both disorders [46,47]. In endometriosis, these cytokines stimulate ectopic endometrial cell proliferation while inducing central sensitization through glial cell activation [48]. ME/CFS studies similarly report TNF-α/IL-6 elevations that correlate with cognitive dysfunction (“brain fog”) and post-exertional malaise [48].

Another critical mechanistic link that may explain the overlapping symptomatology observed in both endometriosis and ME/CFS is dysfunction of the HPA axis4 [49,50]. In endometriosis, chronic pelvic pain and inflammation drive abnormal cortisol secretion patterns, typically characterized by blunted diurnal variation with elevated evening cortisol and reduced morning peaks [51]. This contrasts with the more flattened cortisol profile seen in ME/CFS, where patients frequently exhibit low 24-h urinary free cortisol and attenuated cortisol awakening responses [52]. The HPA axis dysfunction observed in endometriosis—characterized by altered cortisol rhythms—may further predispose patients to ME/CFS by exacerbating fatigue and stress responses [53]. Additionally, the high prevalence of central sensitization in both disorders could reflect overlapping neuroimmune mechanisms, where persistent nociceptive signaling in endometriosis perpetuates the widespread pain and post-exertional malaise central to ME/CFS [54,55].

While there are various mechanisms that could explain the increased association of ME/CFS in patients with endometriosis, our analysis substantiates these findings. Notably, the 17% prevalence of ME/CFS in patients with endometriosis substantially exceeds general population estimates, reinforcing clinical observations of disproportionate fatigue in this population. Notably, we found a bidirectional association between ME/CFS and endometriosis. However, these findings advocate for heightened clinical vigilance using standard diagnostic criteria and testing; fatigue in patients with endometriosis should prompt ME/CFS screening, while patients with ME/CFS reporting pelvic pain may warrant gynecologic evaluation. Therapeutically, interventions targeting shared pathways—such as anti-inflammatory diets or stress-reduction therapies—may benefit both conditions. However, the high heterogeneity in prevalence estimates underscores the need for standardized diagnostic approaches in future studies.

This review’s strengths include rigorous PRISMA adherence, triple-reviewer screening, and quantitative synthesis of associations between endometriosis and ME/CFS.

The trim-and-fill method assumes funnel plot asymmetry and thus suggests publication bias, but other factors (clinical heterogeneity, methodological differences) may contribute. However, the adjusted estimate remains within the original confidence interval, supporting the robustness of our primary findings. The funnel plot asymmetry may reflect selective publication of statistically significant results, underreporting of null or negative findings, or language bias (exclusion of non-English studies). While the trim-and-fill analysis suggests possible publication bias, the clinical significance of this adjustment appears limited, given that both original and adjusted estimates indicate a substantially higher prevalence of ME/CFS in patients with endometriosis compared to general population estimates (typically 0.2–0.7%) [1].

Nevertheless, the limitations of this meta-analysis include (1) a predominance of cross-sectional designs precluding causal inference, (2) potential confounding by unmeasured variables (e.g., anxiety, depression, chronic pain syndrome, autoimmune comorbidities), and (3) overrepresentation of U.S.-based studies, limiting generalizability. In addition, the extreme heterogeneity in some analyses (I^2^ > 98%) suggests pooled estimates should be interpreted cautiously. Moreover, the diagnostic criteria for ME/CFS used for the studies in this analysis varied widely, with almost no study using the commonly accepted CCC [20], ICC [38], or IOM criteria [21]. Even though there are more than 25 case definitions for ME/CFS1, only two studies in our analysis used the Fukuda criteria [19], and one each used the ICD-9 [27] and the CDC-SI [37], while the remaining nine studies documented “self-reported” or “clinical diagnosis/evaluation” as their criteria. Our findings warrant caution, as many studies relied on self-reported diagnoses. More so, it should be recognized that many patients with a self-reported diagnosis of ME/CFS might have had some form of fatigue, as described by the European Network on Myalgic Encephalomyelitis/Chronic Fatigue Syndrome (EUROMENE) group [56].

Thus, we recommend that longitudinal studies, especially prospective cohort or case–control studies using gold standard diagnostic testing (e.g., laparoscopy with biopsy for endometriosis, and the ICC [40] or IOM [21] criteria for ME/CFS) should be conducted to clarify temporal relationships. Future studies must prioritize gold standard criteria to improve diagnostic accuracy and ensure comparability. Until then, the observed associations should be interpreted carefully, as they may reflect methodological artifacts rather than true biological links. Standardization is particularly urgent given emerging evidence that ME/CFS subtypes may correlate with distinct inflammatory profiles, which could differentially overlap with endometriosis pathophysiology. Mechanistic research should explore specific markers as a more robust measure to diagnose ME/CFS. Our findings also underscore the importance of including unpublished data in future systematic reviews on this topic to preclude any potential publication bias.

## 5. Conclusions

This systematic review and meta-analysis demonstrates a significant bidirectional association between endometriosis and ME/CFS, with endometriosis patients exhibiting 2.79-fold higher odds of ME/CFS compared to controls, and ME/CFS patients having 2.52-fold odds of having endometriosis compared to controls. The pooled prevalence of ME/CFS in endometriosis patients was 17%, while the pooled prevalence of endometriosis in ME/CFS cohorts was 13%, though with substantial heterogeneity (I^2^ > 98%). These findings suggest shared pathophysiological mechanisms, potentially involving chronic inflammation, immune dysregulation, and HPA axis dysfunction. However, the extreme variability in diagnostic criteria across studies—particularly the reliance on self-reported rather than standardized clinical diagnoses—limits the certainty of these estimates. The high heterogeneity and risk of bias in cohort studies further underscore the need for rigorous, prospective investigations using validated diagnostic criteria (e.g., laparoscopy for endometriosis and IOM/ICC criteria for ME/CFS).

## Figures and Tables

**Figure 1 diagnostics-15-02332-f001:**
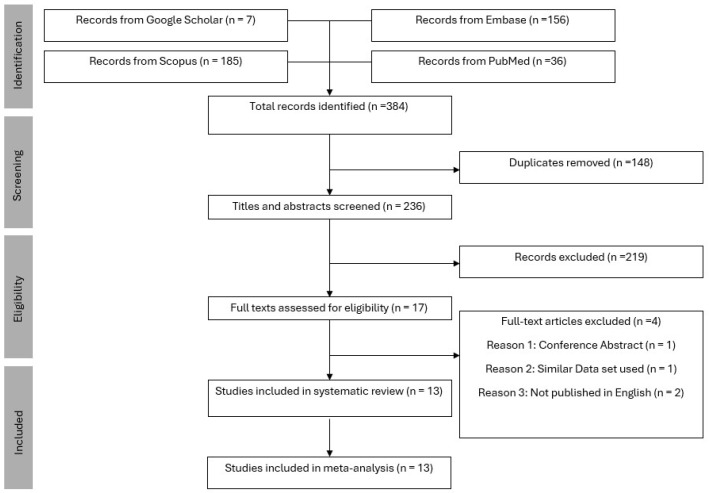
PRISMA flow chart.

**Figure 2 diagnostics-15-02332-f002:**
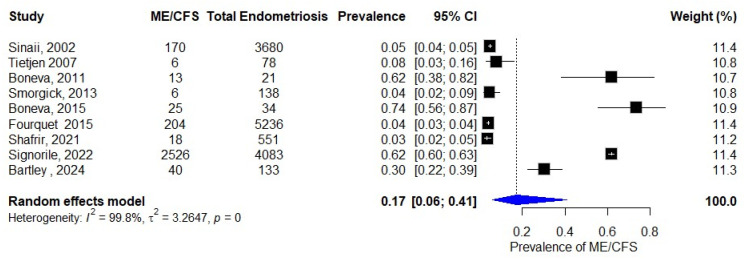
Prevalence of ME/CFS in patients with endometriosis [23,24,25,26,29,30,31,32,35].

**Figure 3 diagnostics-15-02332-f003:**
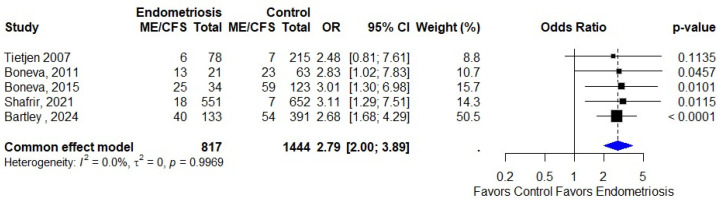
Association of endometriosis and ME/CFS [24,25,29,31,35].

**Figure 4 diagnostics-15-02332-f004:**
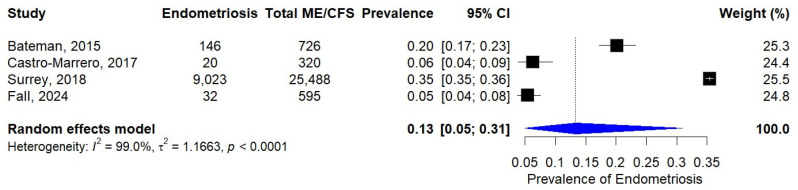
Prevalence of endometriosis in patients with ME/CFS [28,33,34,36].

**Figure 5 diagnostics-15-02332-f005:**
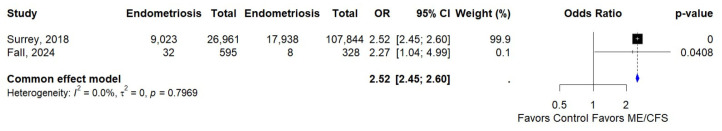
Association of ME/CFS and endometriosis [34,36].

**Figure 6 diagnostics-15-02332-f006:**
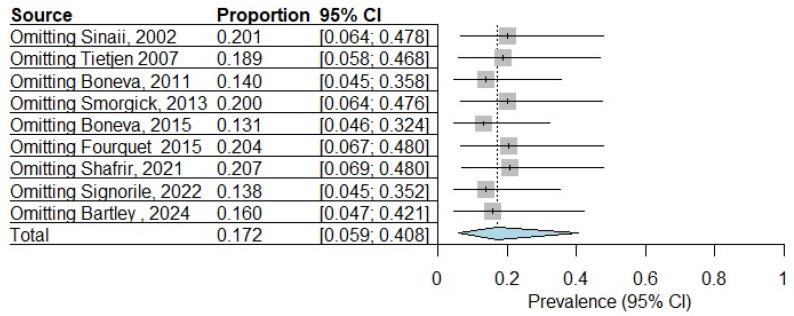
Sensitivity analysis: prevalence of ME/CFS in patients with endometriosis [23,24,25,26,29,30,31,32,35].

**Figure 7 diagnostics-15-02332-f007:**
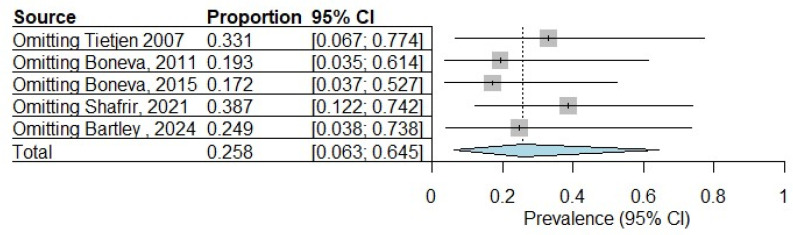
Sensitivity analysis: prevalence of endometriosis in patients with ME/CFS [24,25,29,31,35].

**Figure 8 diagnostics-15-02332-f008:**
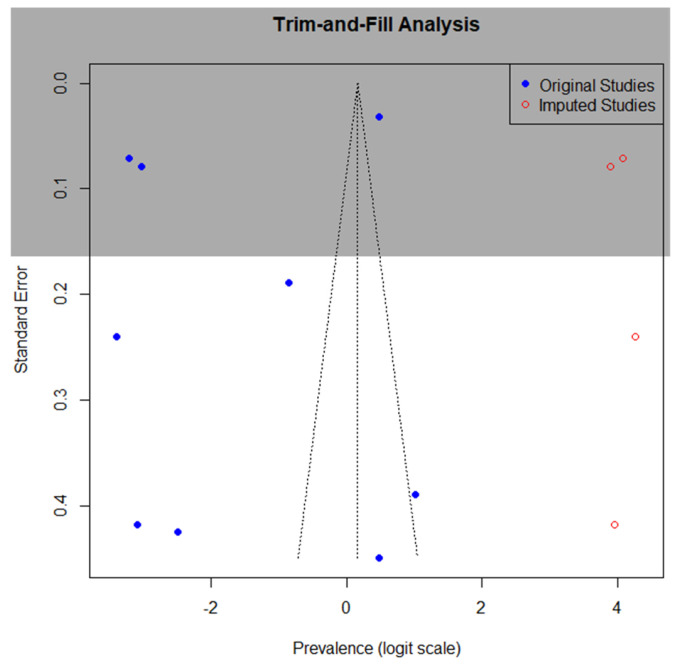
Funnel plot for prevalence of ME/CFS in patients with endometriosis.

**Figure 9 diagnostics-15-02332-f009:**
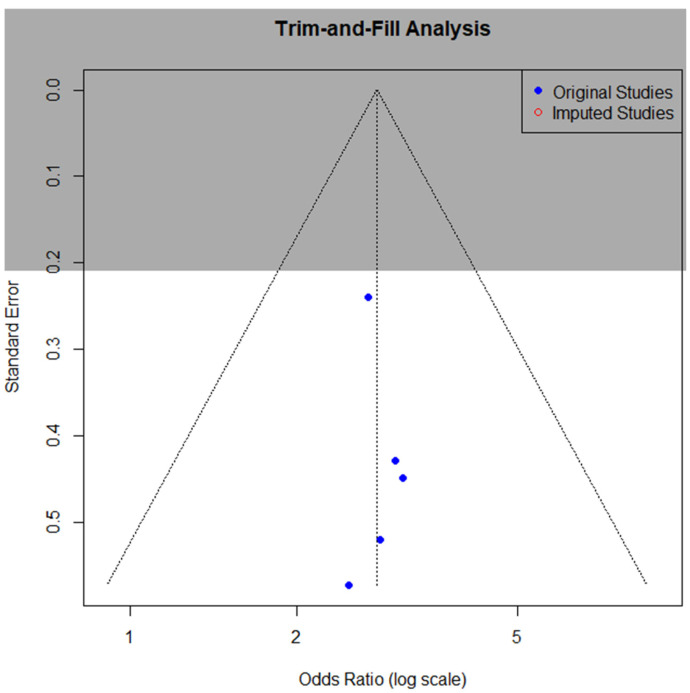
Funnel plot for the association of endometriosis and ME/CFS.

**Figure 10 diagnostics-15-02332-f010:**
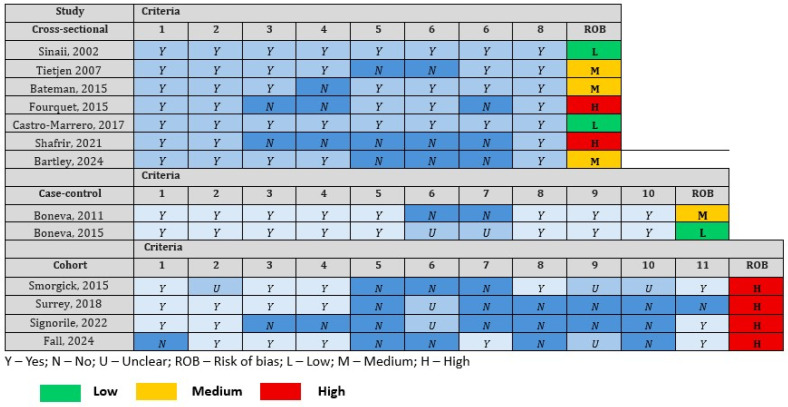
Risk of bias assessment (JBI critical appraisal tool) [23,24,25,26,28,29,30,31,32,33,34,35,36].

**Table 1 diagnostics-15-02332-t001:** Characteristics of included studies.

Study	Country	Study Design	Criteria for Endometriosis	Criteria for ME/CFS	Sample Size
ME/CFS in Patients with Endometriosis					
Sinaii, 2002 [23]	USA	Cross-sectional	Surgical diagnosis	Physician diagnosis	3680
Tietjen 2007 [24]	USA	Cross-sectional	Self-reported	Self-reported	275
Boneva, 2011 [25]	USA	Case–control	Self-reported	Fukuda 1994 [19]	84
Smorgick, 2013 [26]	USA	Retrospective cohort	Surgical diagnosis	ICD-9-CM [27]	138
Bateman, 2015 [28]	USA	Cross-sectional	Self-reported	Fukuda 1994 [19]	960
Boneva, 2015 [29]	USA	Case–control	Self-reported	Fukuda 1994 [19]	157
Fourquet, 2015 [30]	USA/PR	Cross-sectional	Self-reported	Self-reported	4358
Shafrir, 2021 [31]	USA	Cross-sectional	Surgical diagnosis	Self-reported	1203
Signorile, 2022 [32]	Italy	Retrospective cohort	Self-reported	Physical exam, MRI	4083
Endometriosis in Patients with ME/CFS					
Castro-Marrero, 2017 [33]	Spain	Cross-sectional	Surgical	Fukuda 1994 [19]	320
Surrey, 2018 [34]	USA	Retrospective cohort	Clinically diagnosed	Clinical diagnosis	134,805
Bartley, 2024 [35]	USA	Cross-sectional	Self-reported	Self-reported	525
Fall, 2024 [36]	USA	Cross-sectional	Self-reported	CDC-SI [37]	923

ME/CFS—Myalgic encephalomyelitis/chronic fatigue syndrome; ICD-9-CM—International Classification of Diseases, 9th Revision, Clinical Modification; CDC-SI—Centers for Disease Control and Prevention Symptom Inventory; USA—United States of America; PR—Puerto Rico.

## Data Availability

This study used data from previously published studies. Data can be shared upon request.

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
