# Peer review of "Endometriosis and Myalgic Encephalomyelitis/Chronic Fatigue Syndrome: A Systematic Review and Meta-Analysis"

_diagnostics, 2025, doi:10.3390/diagnostics15182332_

Round 1

Reviewer 1 Report

Comments and Suggestions for Authors

thanks for nice study. 

  1. authors may consider consider adding “A PRISMA Systematic Review and Meta-analysis” to aid indexing.
  2. In the introduction part, there are duplicate mechanistic paragraphs (lines 344-370 vs 352-370) 
  3. the author may addresses the thesis to prevent publication-bias risk and other databases (Embase and Web of Science Core, MedRxiv, thesis repositories)
  4. is a priori primary endpoint pooled OR or prevalence? also state software versions for metafor and R fully.

  5. In the results, heterogeneity: I² = 99 % for two prevalence outcomes, however pooled estimates still highlighted in Abstract and Conclusions, such I² indicates the pooled mean is meaningless. I suggest authors to (i) re-analyse using Hartung–Knapp or leave-one-out to identify outliers, and (ii) consider presenting median range rather than pooled prevalence.
  6. I think QUADAS-2 is unsuitable for observational prevalence/incidence studies.these tool mismatches underestimate selection and confounding bias. I suggest to use ROBINS-I (for non-randomised exposure studies) or JBI prevalence tool
  7.  prefer to list specific ME/CFS criteria used (Fukuda, IOM, CCC) instead of “self-report/clinical”. In forest-plots, include weight (%) and in figure legends, add sample units (n).
  8. In the discussion part, please differentiate clearly between association and causation. In the text , it is meant causal risk (e.g., “risk factor-disease model”, line 343).
  9. you may add limitations as lack of control for confounders (e.g., parity, obesity, psychiatric comorbidities).

Author Response

Reviewer 1

Comments 1: thanks for nice study.

Response 1: Dear Reviewer, thank you for kindly providing your insightful comments to make our manuscript scientifically valid. We have looked at each comment carefully and have modified our manuscript accordingly.

Comments 2: authors may consider consider adding “A PRISMA Systematic Review and Meta-analysis” to aid indexing.

Response 2: Thank you for this suggestion. We have added PRISMA as a keyword, to aid indexing.

Comments 3: In the introduction part, there are duplicate mechanistic paragraphs (lines 344-370 vs 352-370)

Response 3: Thank you for this observation. We have looked at the discussion section (the line numbers you indicated, and noted that even though the mechanisms are similar, they deal with the immunological link (lines 344-370) and the hormonal link 352-370).

Comments 4: the author may addresses the thesis to prevent publication-bias risk and other databases (Embase and Web of Science Core, MedRxiv, thesis repositories)

Response 4: Thank you for this suggestion. We have included our search on: Open Access Theses and Dissertations (https://oatd.org/),  and the preprint server for health sciences (https://www.medrxiv.org/)

Comments 5: is a priori primary endpoint pooled OR or prevalence? also state software versions for metafor and R fully.

Response 5: Our main was to look for the association between ME/CFS and endometriosis. During data extraction, we obtained data for a potential bidirectional association. In addition, some studies did not have a control group, therefore it was possible to only estimate the prevalence of one disease in the other.

We have updated our methods regarding the statistical analysis and package used.

Comments 6: In the results, heterogeneity: I² = 99 % for two prevalence outcomes, however pooled estimates still highlighted in Abstract and Conclusions, such I² indicates the pooled mean is meaningless. I suggest authors to (i) re-analyse using Hartung–Knapp or leave-one-out to identify outliers, and (ii) consider presenting median range rather than pooled prevalence.

Response 6: Thank you for this most useful comment. We have performed sensitivity analyses based on the high heterogeneity.

Comments 7: I think QUADAS-2 is unsuitable for observational prevalence/incidence studies.these tool mismatches underestimate selection and confounding bias. I suggest to use ROBINS-I (for non-randomised exposure studies) or JBI prevalence tool

Response 7: Thank you for this suggestion. We agree with you and have redone our risk of bias assessment using the JBI tool, for cross-sectional, case-control, and cohort studies.

 Comments 8: prefer to list specific ME/CFS criteria used (Fukuda, IOM, CCC) instead of “self-report/clinical”. In forest-plots, include weight (%) and in figure legends, add sample units (n).

Response 8: Unfortunately, most of the included studies stated that the diseases were self-reported. However, we have discussed this in the limitations.

Comments 9: In the discussion part, please differentiate clearly between association and causation. In the text , it is meant causal risk (e.g., “risk factor-disease model”, line 343).

Response 9: Thank you for pointing this out. We have changed it to association.

Comments 10: you may add limitations as lack of control for confounders (e.g., parity, obesity, psychiatric comorbidities).

Response 10: Thank you for this recommendation. We have added this in the limitations.

Reviewer 2 Report

Comments and Suggestions for Authors

The present systematic review and meta-analysis addresses a relevant and understudied area regarding the epidemiological and pathophysiological links between endometriosis and myalgic encephalomyelitis/chronic fatigue syndrome (ME/CFS). The methodological approach follows PRISMA standards and PROSPERO registration enhances transparency. However, the following improvements are recommended:

  1. Introduction: Consider clarifying the research gap and hypothesis more explicitly. The introduction is comprehensive but could be condensed for better readability.

  2. Methods: While the systematic search and analysis are appropriate, reliance on self-reported diagnoses for both endometriosis and ME/CFS in many studies weakens the strength of evidence. Please acknowledge this more explicitly as a major limitation.

  3. Results: The high heterogeneity in prevalence estimates and the implausible confidence interval (0 to >1 million) for the reverse association analysis (ME/CFS → endometriosis) should be critically discussed. It is advisable to reconsider how these findings are presented to avoid misleading interpretations.

  4. Discussion: The shared immunological and neuroendocrine pathways are well described, but the discussion would benefit from more emphasis on the diagnostic limitations and potential misclassification bias inherent to included studies. Also, consider softening statements implying a "bidirectional" association, given that statistical significance was not achieved in both directions.

  5. Language: Several sections, particularly in the discussion, are wordy and could be rephrased for clarity and conciseness.

Comments on the Quality of English Language

The text is generally readable, but the wording could be made more concise and clearer, particularly in the discussion section, where some paragraphs are overly long and convoluted.

Author Response

Reviewer 2

Comments 1: The present systematic review and meta-analysis addresses a relevant and understudied area regarding the epidemiological and pathophysiological links between endometriosis and myalgic encephalomyelitis/chronic fatigue syndrome (ME/CFS). The methodological approach follows PRISMA standards and PROSPERO registration enhances transparency. However, the following improvements are recommended:

Response 1: Dear Reviewer, We are grateful for your critique of our manuscript and for providing vital recommendations to increase scientific rigor of our work.

We have modified our manuscript and study methods accordingly, revalidating our data and performing the analyses again. Our work is highly accurate now.

Comments 2: Introduction: Consider clarifying the research gap and hypothesis more explicitly. The introduction is comprehensive but could be condensed for better readability.

Response 2: Thank you for this suggestion. We have updated our introduction.

Comments 3: Methods: While the systematic search and analysis are appropriate, reliance on self-reported diagnoses for both endometriosis and ME/CFS in many studies weakens the strength of evidence. Please acknowledge this more explicitly as a major limitation.

Response 3: Thank you for this comment. Most of the included studies stated that the diseases were self-reported. However, we have discussed this in the limitations.

Comments 4: Results: The high heterogeneity in prevalence estimates and the implausible confidence interval (0 to >1 million) for the reverse association analysis (ME/CFS → endometriosis) should be critically discussed. It is advisable to reconsider how these findings are presented to avoid misleading interpretations.

Response 4: Thank you for catching this error. We have identified our error in our data and have corrected it along with the analysis.

Comments 5: Discussion: The shared immunological and neuroendocrine pathways are well described, but the discussion would benefit from more emphasis on the diagnostic limitations and potential misclassification bias inherent to included studies. Also, consider softening statements implying a "bidirectional" association, given that statistical significance was not achieved in both directions.

Response 5: Thank you for this encouraging feedback and suggestion. We have updated our data extraction and recognized an error. This has been fixed, and our results show a bidirectional association, which is statistically significant. We have also elaborated about the diagnostic limitations in the discussion.

Comments 6: Language: Several sections, particularly in the discussion, are wordy and could be rephrased for clarity and conciseness.

Response 6: Thank you for this comment. We had originally submitted a manuscript with approximately 2600 words, but the journal requirement had a minimum word count of 4000, so we had to change our original manuscript to increase the word count. However, we have tried our best to avoid using convoluted statements while still meeting the journal’s word count requirement.

Round 2

Reviewer 1 Report

Comments and Suggestions for Authors

Author addresses all issues and now the article is ready for publication.

Author Response

I checked this paper which addresses an important issue in clinical practice, that of the association of ME/CFS and endometriosis (EM).

Response:

Dear Editor,

We are grateful for the comments you have provided to make this manuscript scientifically sound. We have duly noted all the comments and made the changes in the best way we can.

As previously discussed, the rationale for linking EM/CFS and EM based on the existence of chronic systemic inflammation in EM should be improved. The assertion "Like ME/CFS, endometriosis involves immune dysregulation, elevated pro-inflammatory cytokines [e.g., interleukin-6 (IL-6), tumor necrosis factor-alpha (TNF-α)], and central sensitization, which may perpetuate chronic pain and fatigue [10–12]" (lines 56-58) seems a little bit insufficient.

Response:

We have rewritten and elaborated on the shared mechanisms, citing different studies.:

The proposed link between endometriosis and ME/CFS is rooted in a cascade of shared biological mechanisms that create a vicious, self-reinforcing cycle. It begins with the im-mune dysregulation inherent in endometriosis, where lesions outside the uterus act as chronic inflammatory sites, flooding the system with pro-inflammatory cytokines like IL-6 and TNF-α. This creates a state of persistent, low-grade systemic inflammation that serves as a constant biological stressor. This inflammatory barrage directly contributes to HPA axis dysfunction, effectively exhausting the body's primary stress-response system and leading to dysregulated cortisol patterns that further cripple the body's ability to quell in-flammation and maintain energy homeostasis.

Concurrently, the unrelenting pain signals from endometriosis lesions promote cen-tral sensitization, a pathophysiological process where the central nervous system becomes hyper-reactive, amplifying pain signals and contributing to the widespread hyperalgesia and fatigue common to both conditions. This neurological rewiring, combined with the constant physiological stress of pain and inflammation, can precipitate autonomic nerv-ous system dysfunction (dysautonomia), manifesting as symptoms like orthostatic intol-erance, palpitations, and gastrointestinal distress—hallmarks of ME/CFS. Thus, the in-flammatory, endocrine, and neurological disruptions caused by endometriosis are not merely comorbid with ME/CFS; they create a perfect storm of pathophysiological overlap that can lower the threshold for developing the full, debilitating syndrome of ME/CFS, ex-plaining the high rate of co-occurrence.

Reference 12 focuses on the role of IL-17A in the context of an in vitro experimental model and could be regarded as more or less inappropriate.

Response:

We have changed and added new appropriate references.

A recent paper deals with the role of proinflammatory cytokines in EM (J Cell Mol Med 2025 Apr 10; 29 (7): e70532) and could provide additional arguments even if the topic seems still debated. Anyway, chronic fatigue is a symptom frequently observed among EM patients (see the recent study Reprod Fertil. 2025 May 15; 6(2):e250010).

Response:

Thank you for these suggestions. We have incorporated them into our manuscript.

I have a concern about the diagnosis criteria used for the selection of included studies. The authors indicated in the Material & Method section that " they employed the PECOS (Population, Exposure, Comparator, Outcomes, Study Design) framework to define study eligibility: Population:Studies involving human participants diagnosed with endometriosis (surgically or clinically confirmed) and ME/CFS (based on recognized criteria such as Fukuda 1994[16], Canadian Consensus Criteria (CCC)[17], or Institute of Medicine (IOM) 2015 criteria[18]; Exposure:Presence of endometriosis (for studies assessing ME/CFS risk) or ME/CFS (for studies assessing endometriosis comorbidity); Comparator:Non-endometriosis controls (for ME/CFS risk studies) or non-ME/CFS controls (for endometriosis studies); Outcomes:Primary outcomes included (1) prevalence of ME/CFS in endometriosis populations (or vice versa), and (2) odd ratios (OR) quantifying association strength; Study Designs: We included observational studies (cohort, case-control, cross-sectional), and clinical trials reporting primary data. Case reports, conference abstracts, letters, review articles, opinion pieces, editorials, and non-peer-reviewed articles were excluded" (lines 104-119). We agree with this approach but in the Results section, authors presented the selected papers (Table 1) which mixed studies using for the EM diagnosis a surgical proof or patient self-reports. Similarly, the use of EM/CFS term should be restricted to the 5 studies including patients diagnosed according to international criteria (Fukuda/CDS, ICC, IOM, NICE...). To avoid any confusion, I recommend to authors to use the terminology proposed by EUROMENE group:  "Chronic fatigue-spectrum disorder (CFSd) is an encompassing term and may be used to refer to persistent profound fatigue for over 3–6 months associated with other symptoms, including the following sub-categories: (a) cases meeting diagnostic criteria for ME/CFS; (b) cases that do not fully meet diagnostic criteria (Non-ME chronic fatigue-Sd) but cannot be explained otherwise; (c) cases totally or partially explained by other diseases known to cause chronic fatigue (disease-associated CFS; or ME/CFS of combined aetiology) (Medicina 2021, 57, 510).

Response:

Thank you for pointing out these diagnostic inconsistencies and for recommending using the CFSd proposed by the EUROMENE group.

While it is an excellent recommendation to use the CFSd consensus criteria, we believe that using it will deviate from our focus. Moreover, a search on PubMed for “Chronic fatigue-spectrum disorder” yielded only 6 results (with most results relating to ME/CFS), and a search on Google producing similar findings, with no article on CFSd per se. However, we included self-reported diagnosis for endometriosis and ME/CFS in our populations, in addition to including it in our limitations.

2.3 Study Selection and Eligibility Criteria

We employed the PECOS (Population, Exposure, Comparator, Outcomes, Study Design) framework to define study eligibility:

Population: Studies involving human participants diagnosed with endometriosis (either surgically or clinically confirmed or from self-report) and ME/CFS (based on recognized criteria such as Fukuda 1994[19], Canadian Consensus Criteria (CCC)[20], or Institute of Medicine (IOM) 2015 criteria[21], or from self-reported data).

Discussion

Our findings warrant caution, as many studies relied on self-reported diagnoses. Moreso, it should be recognized that many patients with self-reported diagnosis of ME/CFS might have had some form of fatigue, as described by the European Network on Myalgic Encephalomyelitis/Chronic Fatigue Syndrome (EUROMENE) group[56].

There are mistakes in the numbering of the paragraphs and numerous grammatical typos that must be corrected. I hope that these remarks may be useful.

Response:

Thank you for noticing these errors. We have numbered the paragraphs correctly. In addition, we have used Grammarly and Microsoft Word Editor to correct any spelling or grammatical errors.